# Effects of cytochrome P450 2B6 and constitutive androstane receptor genetic variation on Efavirenz plasma concentrations among HIV patients in Kenya

Musa Otieno Ngayo[1,2]*, Margaret Oluka[2], Zachari Arochi Kwena[1], Wallace Dimbuson Bulimo[3], Faith Apolot Okalebo[2]

1 Centre for Microbiology Research, Kenya Medical Research Institute, Nairobi, Kenya, 2 Department of Pharmacology and Pharmacognosy, School of Pharmacy, University of Nairobi, Nairobi, Kenya, 3 Department of Biochemistry, School of Biological and Physical Sciences, University of Nairobi, Nairobi, Kenya

* mngayo@kemri.org

**Data Availability Statement:** Ngayo, Musa; Oluka, Margaret; Arochi, Zachari; Bulimo, Wallace Dimbuson; Okalebo, Faith Apolot (2021): Effects of

## Abstract

The effects of genetic variation of cytochrome P450 2B6 (CYP2B6) and constitutive androstane receptor (CAR) on efavirenz (EFV) plasma concentration was evaluated among 312 HIV patients in Nairobi Kenya. The EFV plasma concentration at steady-state were determined using ultra-high-performance liquid chromatography with a tandem quadruple mass spectrometer (LC-MS/MS). Thirteen CYP2B6 (329G>T, 341T>C, 444 G>T/C, 15582C>T, 516G>T, 548T>G, 637T>C, 785A>G, 18492C>T, 835G>C, 1459C>T and 21563C>T) and one CAR (540C>T) single nucleotide polymorphisms (SNPs) were genotyped using real-time polymerase chain reaction. HIV drug resistance mutations were detected using an in-house genotypic assay. The EFV concentration of patients ranged from 4 ng/mL to 332697 ng/mL (median 2739.5 ng/mL, IQR 1878–4891.5 ng/mL). Overall, 22% patients had EFV concentrations beyond therapeutic range of 1000–4000 ng/mL (4.5%% < 1000 ng/mL and 31.7% > 4000 ng/mL). Five SNPs (15582C>T, 516G>T, 785A>G, 983T>C and 21563C>T) were associated with higher EFV plasma concentration while 18492C>T with lower EFV plasma concentration (p<0.05). Strong linkage disequilibrium (LD) was observed for 15582C>T, 516G>T, 785A>G, 18492C>T, 983T>C, 21563C>T, 1459C>T and CAR 540C>T. Sixteen haplotypes were observed and CTGCTTCC, CTGCTTCT, TTGCTTCT and CGACCCCT were associated with high EFV plasma concentration. In multivariate analysis, factors significantly associated with EFV plasma concentration included; the presence of skin rash (β = 1379, 95% confidence interval (CI) = 3216.9–3416.3; $p < 0.039$), T allele of CYP2B6 516G>T (β = 1868.9, 95% CI 3216.9–3416.3; $p < 0.018$), the C allele of CYP2B6 983T>C (β = 2638.3, 95% CI = 1348–3929; $p < 0.0001$), T allele of CYP2B6 21563C>T (β = 1737, 95% CI = 972.2–2681.9; $p < 0.0001$) and the presence of 5 to 7 numbers of SNPs per patient (β = 570, 95% CI = 362–778; $p < 0.0001$) and HIV viral load $\leq$1000 cells/mL (β = -4199.3, 95% CI = -7914.9 —-483.6; $p = 0.027$). About 36.2% of the patients had EFV plasma concentrations beyond therapeutic window, posing high risk of treatment failure or toxicity. The SNPs of CYP2B6 516G>T, CYP2B6 983T>C, 21563C>T, presence of higher numbers

cytochrome P450 2B6 and constitutive androstane receptor genetic variation on Efavirenz plasma concentrations among HIV patients in Kenya. figshare. Dataset. https://doi.org/10.6084/m9.figshare.15026289.v1.

**Funding:** This study was supported by funds from KEMRI-Internal Grant (IRG/20) 2010/2011.

**Competing interests:** The authors have declared that no competing interests exist.

of SNPs per patient and haplotypes `CTGCTTCC`, `CTGCTTCT`, `TTGCTTCT` and `CGACCCCT` could efficiently serves as genetic markers for EFV plasma concentration and could guide personalization of EFV based ART treatment in Kenya.

## Introduction

Efavirenz (EFV) is primarily metabolized to 8-hydroxyefavirenz by cytochrome P450 2B6 (CYP2B6) [1] and to a lesser extent to 7-hydroxy-EFV by CYP2A6 [2,3]. The direct N-glucuronidation of EFV metabolites for excretion by UDP-glucuronosyltransferase (UGT) isoforms (including UGT1A1 and 2B7) represent minor metabolic pathway [4,5]. The transcription factors pregnane X receptor (PXR, *NR1I2*) and constitutive androstane receptor (CAR, *NR1I3*) act on genes involved in xenobiotic metabolism and excretion [6–8]. EFV has the ability to autoinduce its own metabolism through the activation of PXR and CAR [7,9].

High genetic polymorphism has been observed in the CYP2B6 gene with several non-synonymous, synonymous and promoter SNPs identified [10]. Currently, about 38 CYP2B6 alleles (*1A [wild-type] to *38) associated with either increased, decreased or abolished enzymatic activity have been defined [11,12]. A number of CYP2B6 SNPs influencing EFV plasma levels such as 516G>T, 785A>G, 983T>C, and 1459C>T have been studied in details [13–15]. Nonetheless, investigating an individual SNP often do not offer satisfactory data predicting inter and intra personal variations in EFV plasma levels. To provide a more accurate data on the influence of SNPs on EFV plasma levels, studies are recommending evaluating a battery of SNPs that reduce the metabolic function of CYP2B6 [16]. This study evaluated the effect of thirteen CYP2B6 (329G>T, 341T>C, 444 G>T/C, 15582C>T, 516G>T, 548T>G, 637T>C, 785A>G, 18492C>T, 835G>C, 1459C>T and 21563C>T) and one CAR (540C>T) single nucleotide polymorphisms (SNPs) on efavirenz plasma concentrations. Additionally, the association of CYP2B6 and CAR polymorphisms and haplotypes with efavirenz plasma concentrations were also investigated.

## Materials and methods

### Study design and setting

This cross-sectional study was conducted between August, 2018 to January 2020. Consenting and enrollment was done for 312 HIV patients receiving HIV care and treatment at the Family AIDS Care and Educational Services (FACES) based at Kenya Medical Research Institute (KEMRI), Nairobi Kenya. Data presenting in this study was part of a study that aimed at evaluating the pharmacogenetic and pharmacoecologic etiology of sub-optimal responses to non-nucleoside reverse transcriptase inhibitor (NNRTI) for the purpose of individualization of HIV treatment in Kenya. Other than receiving ART treatment in FACES-KEMRI, patients were recruited in this study if they were: (i) aged above 18 year (ii) consenting to the study, (iii) be on ARV treatment for 12 months, and (iv) had been receiving first line ARV (Zidovudine (AZT) or Abacavir (ABC), 3TC, and EFV. The detail of this cohort has been described in detail in our previous publication [17].

### Ethical statement

This study was done according to the principles of the Declaration of Helsinki and was approved by the KEMRI Scientific Review Unit (SERU) (SSC No. 2539). Before recruitment in this study, all patients filled in a written informed consent for study participation.

## Data collection

A detailed, structured, face-to-face interview gathered information on patient's socio-demographic, ARV use and medical history. Blood samples (10 mL) at 12–16 h post ARV uptake were collected into three blood tube as follows: EDTA anticoagulant tube for immunological testing and CYP2B6 and CAR genotyping. Serum separating tube for clinical chemistry while Lithium heparin tube for HIV viral load and EFV plasma level quantification. The samples were stored at −80˚C after collection awaiting analysis.

## Quantification of EFV plasma concentrations

**Solutions.** Efavirenz (purity: 99.0%) and internal standard C6-efavirenz (purity: 99.3%) were purchased from Alsachim (Strasbourg, France). The 200 μg/ml EFV stock solution was diluted with 50% methanol in water to concentration ranges of 523.56 to 62000.00 ng/ml. The internal standard was diluted in 50% methanol to give a working solution of 100 ng/mL Then 20μL working standard and 20μL IS was further diluted in 200μL drug free human plasma to prepare 6 plasma calibrators at 10-fold dilution.

**Selectivity.** The selectivity of endogenous plasma constituents was evaluated using six different sets of plasma samples by analyzing blanks and spiked samples at Low quality control (LQC) levels. The EFV in the plasma spiked at the LQC level and clinical samples was detected at its retention time with single peak an indication that the method was selective to EFV.

**Method recovery and linearity.** The data for absolute recovery of EFV for six replicates at Low quality control (LQC), middle quality control (MQC) and high-quality control (HQC) level were higher than 80% recovery, further showing the suitability of the method to analyze these two drugs.

**Method accuracy and precision.** Intra-day and inter-day accuracy and precision was evaluated at three different concentrations (LQC, MQC and HQC) for EFV. For inter-batch, three runs and for intra-batch, a single run was assayed. Each run consisted of six replicates. Both the intra- and inter- day accuracy and precision values were within the acceptance ranges. For EFV, intra-day accuracy ranged from 92.1% to 102% with an inter- day accuracy of 96.7% to 101.4%. The EFV intra-day precision ranged from 5.1% to 7.7% with an inter-day variation of 6.9% to 9.2%.

**Viral inactivation.** The quantification was achieved first by the inactivation of HIV virus as follows. The 50μl of plasma of each sample and 5μl internal standard in a 1.5ml Eppendorf tube was heated at 65˚C for 10 minutes and subsequently cooled at room temperature for 10 minutes. A 100μl cold methanol (-20˚C) was then added and kept at -20˚C for 10 minutes. The samples were then centrifuged at 20,000g at 20˚C for 8 minutes and the supernatant collected in a clean 1.5ml Eppendorf tube. The 850μl ammonium acetate buffer (pH = 3.00) was added to the supernatant and briefly centrifuged. The sample was considered safe to be handled in a non P3 laboratory.

**Solid phase extraction using C18 Cartridge and quantification of EFV.** The EFV plasma concentrations were measured using a tandem quadrupole mass spectrometer designed for ultra-high performance: Xevo TQ-S (Waters Corporation, U.S.A) as described by Reddy *et al.* [18]. The Bond Elute C18 cartridges were prepared and placed onto the Visiprep Vacuum Manifold with Standard Lid (Merck, Germany). The Bond Elute C18 150 × 4.6 mm, 5-μm column was conditioned by first passing through 1 ml methanol followed by 1 ml ultra-pure water. Each column was then charged with 150μl samples containing 850μl ammonium acetate buffer (pH = 3.00) followed by twice cleaning using 1 mL ultrapure water. The first cleaning was collected into clean separate tube while the second water cleaning collected in the waste tubes. The columns were vacuum dried (5–10 kpa in Hg). The efavirenz elution at a flow

rate of 1 ml/min was then done twice using methanol 500μl with vacuum drying between the two elution. Elutes were then completely evaporated using Thermo Scientific™ Reacti-Vap™ Evaporators (Thermo Fisher Scientific Inc, USA) at 37˚C for 30 min. This was then reconstituted using 100μl of equal parts 1:1 acetonitrile and water, vortexed briefly and transferred into 50 ml capped vials and placed into Xevo TQ-S (Waters Corporation, U.S.A) for quantification. About 1μl of the samples was injected automatically into the LC/MS/MS instrument and quantified within 5 minutes. The EFV plasma concentration was categorized as <1000 ng/ml considered below therapeutic range, 1000 to 4000ng/ml considered therapeutic range and >4000 ng/ml considered supratherapeutic level [19,20].

## CYP2B6 and CAR genotyping

**DNA preparation.**  Genomic DNA was extracted from EDTA collected blood using automated NucliSENS® easyMAG® system (BioMérieux—Boston, USA) according to the manufacturer's instructions. The quality of DNA was measured using a ND-1000 UV spectrophotometer (NanoDrop Technologies, Wilmington, DE, USA).

**Real time PCR genotyping.**  Genotyping was carried out on an ABI 7500 Fast Sequence Detection System (Applied Biosystems, Foster City, CA, USA). SNPs were analyzed using the validated Taqman Genotyping Assays for *CYP2B6* 516G>T (rs3745274), CYP2B6 983T>C (rs28399499), CYP2B6 15582C>T (rs2279345), CYP2B6 18492 C>T (rs2279345), CYP2B6 21563 C>T (rs8192719) and CAR 540C>T (rs2307424) applied Biosystem pre-validated assays were utilized. The assay IDs were C___7817765_60, C_60732328_20, C__26823975_10, C__26823975_10, C__22275631_10 and C__25746794_20 respectively. These assays were done according to the manufacturer's instructions. Briefly, in a final volume for each reaction of 20ul, was 2X TaqMan Genotyping Master Mix (Applied Biosystems, Foster City, CA, USA), 20X Drug Metabolism Genotyping Assay and 10ng genomic DNA. The PCR consisted of an initial step at 95˚C for 10 minutes and 50 cycles at 92˚C for 15 seconds and 90˚C for 60 seconds.

The primers for CYP2B6 329 G>T (rs186335453), CYP2B6 341T>C (rs139801276), CYP2B6 444 G>T (rs1053569), CYP2B6 548 T>C (ss539003292), CYP2B6 637 T>C (ss539003292), CYP2B6 785A>G (rs2279343), CYP2B6 835 G>C(rs139029625) and CYP2B6 1459 C>A (ss539003296) are listed in Table 1. These SNPs were also genotyped using an ABI 7500 Fast Sequence Detection System (Applied Biosystems, Foster City, CA, USA) as described by Radloff et al., [21]. Briefly, in a final volume for each reaction of 20ul, was 2X TaqMan Genotyping Master Mix (Applied Biosystems, Foster City, CA, USA), primers forward (10um) and reverse (10 uM), wildtype and mutant probes (10 uM each), $H_2O$ and 10ng genomic DNA. The PCR consisted of an initial step at 50˚C for 2 minutes, 95˚C for 10 minutes and 45 cycles at 95˚C for 15 seconds and 60˚C for 60 seconds. The plates were read using the allelic discrimination settings. The SNP assay was set up using SDS, version 1.3.0 as an absolute quantification assay. Post-assay analysis was done using SDS software. The results for these SNPs were defined as either homozygous wild type, heterozygous and homozygous mutant.

**Blood chemistry.**  The CD4 cell counts were measured using a FACSCount TM flow cytometer (BD Biosciences, San Jose, USA) while HIV-1 RNA was measured using Generic HIV Viral Load® (Biocentric, Bandol, France). These assays were done according to manufactures instructions.

**HIV drug-resistant genotyping.**  The presence of HIV drug-resistant mutation was tested using an in-house genotypic method previously described by Lehman *et al.*, [22]. Resistance mutations were identified using the Stanford University and International AIDS Society-USA website Interpretation Algorithm.

**Table 1. List of CYP2B6 SNPs primers used for genotyping in this study (Radloff et al., 2013) [21].**

| SNP | Region | Variant (rs number) | Direction | Sequence (5'– 3') c |
|---|---|---|---|---|
| 329 G>T | Exon 2 | rs186335453 | F | CGACCCATTCTTCCGGGTATATGGTGTGATCTTTG |
|  |  |  | R | CAAAGATCACACCATATACCCGGAAGAATGGGTCG |
| 341 T>C | Exon 3 | rs139801276 | F | CCGGGGATATGGTGTGACCTTTGCCAATGGAAACC |
|  |  |  | R | GGTTTCCATTGGCAAAGGTCACACCATATCCCCGG |
| 444 G>T | Exon 3 | rs1053569 | F | GGAGCGGATTCAGGATGAGGCTCAGTGTCTG |
|  |  |  | R | CAGACACTGAGCCTCATCCTGAATCCGCTCC |
| 548 T>C | Exon 4 | ss539003292 | F | CATCATCTGCTCCATCGGCTTTGGAAAACGATTCC |
|  |  |  | R | GGAATCGTTTTCCAAAGCCGATGGAGCAGATGATG |
| 637 T>C | Exon 4 | ss539003294 | F | CTTTTTCACTCATCAGCTCTGTACTCGGCCAGCTGT |
|  |  |  | R | ACAGCTGGCCGAGTACAGAGCTGATGAGTGAAAAAG |
| 785 A>G | Exon 5 | rs2279343 | F | AGGCAAGTTTACAAAAACCTG |
|  |  |  | R | CCCTCCCTAGTCTTTCTTCTTCC |
| 835 G>C | Exon 6 | rs139029625 | F | GAAAAAGAGAAATCCAACCCACACAGTGAATTCAGCC |
|  |  |  | R | GGCTGAATTCACTGTGTGGGTTGGATTTCTCTTTTTC |
| 1459 C>A | Exon 9 | ss539003296 | F | CAACATACCAGATCAGCTTCCTGCCCCGC |
|  |  |  | R | GCGGGGCAGGAAGCTGATCTGGTATGTTG |

**Statistical analysis.** Statistical analyses were done using Stata version 13 (StataCorp. LP, College Station, USA). Steady-state efavirenz plasma concentrations were tested for normality by the Shapiro- Wilk test. Evaluation of Hardy-Weinberg equilibrium (estimation of p-values was calculated using the Markov chain method) for the 13 CYP2B6 SNPs and 1 CAR genetic variation, Linkage Disequilibrium, allele, genotype and haplotype frequency and differences in the SNP/allele frequencies between groups/ populations using the SNPStats software -free web tool for SNP analysis [23]. Wright's F statistics was applied to evaluate the expected level of heterozygosity. Variation in log10-transformed EFV plasma levels across clinical and genetic factors were on EFV plasma concentrations was determined using Kruskal-Wallis test and Dunn's test by ranks. Quantile regression analysis was used to evaluate pharmacogenetical factors associated with EFV plasma levels. The significance level was set at $P \leq 0.05$.

## Results

### Baseline characteristics of study participants

A total of 312 patients were evaluated, with a median age of 40 years (interquartile range (IQR) = 34–47 years), 179 (57.4%) were female, 206(66%) were bantus, 60(19.2%) had a previous partner who had died due to HIV infection. Twenty (6.4%) of the patients had skin rash, 18 (5.8%) were smokers while 54(17.3%) were consuming alcohol. The majority of the patients 187(59.9%) were taking lamivudine/efavirenz/ tenofovir based ART regimen with a 207 (66.3%) non-adherence rate in the past 30 days. The median CD4 was 404.5 cell/mL (IQR = 273.5–543.5 cells/mL), with a median body mass index (BMI) of 24.6 kg/m2 (IQR = 21.5–29.2 kg/m2), median ALT of 25UL (IQR = 19–39.5 UL), median AST of 28 UL (IQR = 20–38 UL) (Table 2).

### Efavirenz plasma concentration

The steady-state EFV plasma concentrations varied widely among patients, ranging from 4 ng/ mL to 332697 ng/mL (median 2739.5 ng/mL, IQR 1878–4891.5 ng/mL). The patients on efavirenz plasma concentration were distributed as follows, the majority 199(63.8%) had plasma

**Table 2. Summary of patient demographics and clinical characteristics of patients.**

| Variables | All patients | Efavirenz plasma concentration | | | p value |
|---|---|---|---|---|---|
| | | <1000ng/ml | 1000–4000ng/ml | >4000ng/ml | |
| | | Sub-therapeutic range | Therapeutic range | Above therapeutic range | |
| | | 14 (4.5%) | 199 (63.8%) | 99 (31.7%) | |
| Age (years), Median (IQR) | 40(34–47) | 45.5(40–52) | 40(35–47) | 39(32–46) | 0.083 |
| Gender Female, n (%) | 179(57.4) | 7(50) | 120(60.3) | 52(52.5) | 0.376 |
| Ethnicity, Bantus, n (%) | 206(66.1) | 11(5.3) | 130(63.1) | 65(31.6) | 0.753 |
| Living with partner, n (%) | 199(63.8) | 5(2.5) | 124(62.3) | 70(35.2) | 0.032 |
| Age of sexual debut (Years), Median (IQR) | 18(17–20) | 17(16–20) | 18(17–20) | 18(16–19) | 0.531 |
| Duration with HIV (Years), Median (IQR) | 10(8–13) | 10(8–15) | 10(8–13) | 10(8–14) | 0.216 |
| Skin rash, Yes n (%) | 20(6.4) | 3(15) | 13(65) | 4(20) | 0.045 |
| Duration ART use (Months), Median (IQR) | 24(12–28) | 22(6–27) | 24(12–27) | 24(9–29) | 0.983 |
| Current ARV Type, n (%) | | | | | 0.329 |
| lamivudine, efavirenz, Abacavir | 1 (0.3) | 0 | 1(100) | 0 | |
| lamivudine, efavirenz, tenofovir | 187 (59.9) | 12(6.4) | 116(62) | 59(31.6) | |
| lamivudine, efavirenz, zidovudine | 124 (39.7) | 2(1.6) | 82(66.1) | 40(32.3) | |
| Missed ART scheduled visit, n (%) | 25 (8.1) | 3(12) | 12(48) | 10(40) | 0.079 |
| Non-adherence in the past 30 days, n (%) | 207(66.3) | 8(3.9) | 139(67.2) | 60(28.9) | 0.197 |
| HIV-RNA copies/mL, n (%) | | | | | 0.0001 |
| Failure (>1000copies/mL) | 12 (3.9) | 4(33.3) | 7(58.3) | 1(8.3) | |
| Responders (<1000copies/mL) | 300 (96.1) | 10 (3.3) | 192(64) | 98(32.7) | |
| Presence of HIV drug resistant mutation | 10(3.2) | 3(30) | 6(60) | 1(10) | 0.005 |
| CD4 Cells/mL, Median (IQR) | 404.5(273.5–543.5) | 285.5(126–510) | 408(279–538) | 403(278–563) | 0.77 |
| AST (U/L), Median (IQR) | 28(20–38) | 23(18–34) | 28(20–38) | 28(19.2–38) | 0.688 |
| ALT (U/L), Median (IQR) | 25(19–39.5) | 22(13–36) | 25(19–41) | 26(19–41) | 0.342 |
| BMI (kg/m2), Median (IQR) | 24.6(21.5–29.2) | 23.8(22–26.5) | 24.7(21.3–29) | 24.6(21.7–30.1) | 0.346 |

concentrations between 1000 to 4000 ng/mL considered levels for viral mutant selection windows followed by 99(31.7%) had supra-therapeutic EFV plasma levels (>4000 ng/mL) while 14(4.5%) had plasma concentrations of <1000 ng/ml considered levels for poor viral suppression window and (p <0.05) (Table 2).

Patients with EFV plasma level within therapeutic 1000 and 4000 ng/mL were those living with their partners (62%) compared to patients with EFV plasma level of <1000 ng/mL or >4000 ng/mL (2.5% and 35.2% respectively; p = 0.032). For patients with skin rash 20% had EFV plasma level of >4000 ng/mL compared to 15% of the patients with EFV plasma level of <1000ng/mL; (p = 0.045). Among patients with virologic failure (>1000copies/mL) although the majority (58.3%) had EFV plasma level between 1000 ng/mL-4000ng/mL, a significant number (33.3%) had sub-optimal EFV plasma level of <1000ng/mL compared to 8.3% with supra-therapeutic EFV plasma level >4000ng/mL; (p = 0.0001). Similarly, among the 10 patients who had NNRTI drug resistant mutations, although the majority (60%) had EFV plasma level between 1000 ng/mL-4000ng/mL, a significant number (30%) had sub-optimal EFV plasma level of <1000ng/mL compared to 10% with supra-therapeutic EFV plasma level >4000ng/mL; (p = 0.005). No differences were observed with regards to gender, ART regimen type, non-adherence, median Age, duration with HIV, duration on ART, CD4 count, AST and ALT levels and BMI in patients with supratherapeutic EFV levels > 4000 ng/mL when compared to patients with sub-optimal EFV level. A trend was observed in the different strata with regards to missing ARV scheduled appointment and median age (Table 2).

## Allele and genotype frequencies of CYP2B6 gene and CAR SNPs

The heterozygous or homozygous mutant or both were not detected for five CYP2B6 SNPs including: CYP2B6 329G>T, 341T>C, 444 G>T/, 637T>C, 835G>C, 548T>G and were not analyzed further. Seven CYP2B6 SNPs (516G>T, 785A>G, 983C>TA, 18492C>T, 21563C>T, 1459C>T and 15582 C>T) and one CAR SNP (540C>T) conformed to Hardy–Weinberg equilibrium and were further analyzed. The allele and genotype frequencies of these SNPs and association with EFV concentrations are summarized in Table 3. Kruskal-Wallis test analysis showed that the homozygous mutant for 155882C>T, 516G>T, 785A>G, 983T>C, 21563C>T are associated with significantly high median (IQR) EFV plasma concentration ($p<0.05$). Only SNP 18492C>T had both heterozygous and homozygous mutant significantly associated with lower median (IQR) EFV plasma concentration ($p<0.05$). The median (IQR) plasma EFV levels was not significantly associated with 1459C>T and CAR 540C>T (Table 3 and S1 Fig).

## CYP2B6 and CAR haplotype frequency and association with EFV concentrations

We quantified the extent of LD among the CYP2B6 and CAR SNP pairs among study patients. Strong LD, defined by high values for both D' ($\geq 0.8$) and r2 ($\geq 0.5$) parameters, was only observed between SNP pairs CYP2B6 516–785, 516–21563 and 785–2156. The 516–18492 pair, 785-18492pair and 18492–21563 pair had high D' ($>0.8$) and moderate r2 (0.12–0.351) value. The CAR 540–18492 pair, had high D' ($\geq 0.8$) and moderate r2 (0.155) values. All other SNP pairs had highly variable D' (0–0.8) and low r2 ($<0.1$) values among the patients (S2 Fig). These findings indicate strong linkage disequilibrium among 15582C>T, 516G>T, 785A>G, 18492C>T, 983T>C, 21563C>T, 1459C>T and CAR 540C>T, resulting in 16 haplotypes among which CTGCTTCC was the most common occurring 101(32.4%) with CGGCTTCC and TGATTTCC reported in 1(0.3%) patient each. The haplotypes (CTGCTTCC, CTGCTTCT, TTGCTTCT and CGACCCCT) were associated with higher EFV plasma concentration (Table 4).

## Quartile regression model

In the final multivariate analysis, factors associated with a higher EFV plasma concentration included: presence of skin rash ($\beta$ = 1379, 95% confidence interval (CI) = 3216.9–3416.3;

**Table 3. Allele and genotype frequencies of CYP2B6 gene and CAR SNPs and their relationship with EFV plasma concentrations.**

| SNPs | Allele | Allele Frequency–n (%) | | Genotype Frequency–n (%) | | | Median EFV Concentration (IQR) -μg/mL | | | P value |
|---|---|---|---|---|---|---|---|---|---|---|
| | A1/A2 | A1 | A2 | A1/A1 | A1/A2 | A2/A2 | A1/A1 | A1/A2 | A2/A2 | |
| **CYP2B6 15582C>T** | C/T | 560(0.9) | 64(0.1) | 255(0.82) | 50(0.16) | 7(0.02) | 2747 (1918–5204) | 2402(1633–3519) | 43788(2539–9313) | 0.07 |
| **CYP2B6 516G>T** | G/T | 395(0.63) | 229(0.37) | 128(0.41) | 139(0.45) | 45(0.14) | 2037.5(1501–3170) | 2754(1985–4487) | 8282(504–13564) | 0.0001 |
| **CYP2B6 785A>G** | A/G | 394(0.63) | 230(0.37) | 127(0.41) | 140(0.45) | 45(0.14) | 2043(1548–3182) | 2743(1968–4403) | 8282(504–13564) | 0.0001 |
| **CYP2B6 18492C>T** | C/T | 511(0.82) | 113(0.18) | 207(0.66) | 97(0.31) | 8(0.003) | 3300(2014–6745) | 2059(1718–3095) | 2391(1434–3034) | 0.0001 |
| **CYP2B6 983T>C** | T/C | 578(0.93) | 46(0.07) | 268(0.86) | 42(0.13) | 2(0.01) | 2580(1836–4420) | 3810(2520–10554) | 8523(7572–9473) | 0.002 |
| **CYP2B6 21563C>T** | C/T | 394(0.63) | 230(0.37) | 126(0.4) | 142(0.46) | 44(0.14) | 2038(1548–3182) | 2760(1985–4487) | 7970(4978–13188) | 0.0001 |
| **CYP2B6 1459C>T** | C/T | 621(0.99) | 3(0.005) | 310(0.99) | 1(0.003) | 1(0.003) | 2739(1870–4872) | 332701(332701–332701) | 2019(2019–2019) | 0.19 |
| **CAR 540C>T** | C/T | 574(0.92) | 50(0.08) | 264(0.85) | 46(0.15) | 2(0.01) | 2687(1878–5025) | 3017(1833–4872) | 2640(2019–3262) | 0.966 |

n—number; %—percentage; IQR—Interquartile range; p value—Significant level.

**Table 4. Relationship between haplotypes and EFV plasma concentrations.**

| Haplotypes | n (%) | Efavirenz plasma concentration | | | |
|---|---|---|---|---|---|
| | | <1000ng/ml | 1000–4000 ng/ml | >4000ng/ml | Median (IQR) ng/ml |
| | | 14 (4.5%) | 26 (6.9%) | 255 (67.6%) | |
| CTGCTTCC | 101 (32.4) | 2(1.9) | 43(42.6) | 56(55.5) | 3001 (1925–5773) |
| CGATTCCC | 52 (16.7) | 2(3.9) | 45(86.5) | 5(9.6) | 2529.5(1903–4063) |
| CGACTCCC | 35 (11.2) | 1 (2.9) | 28(80) | 6(17.1) | 1991(1252–3095) |
| CTGTTTCC | 35 (11.2) | 2(5.7) | 28(80) | 5(9.6) | 2368(1740–4713) |
| TGACTCCC | 25 (8) | 2(8) | 18(72) | 5(20) | 3572(2526–6116) |
| TTGCTTCC | 22 (7.1) | 1(4.5) | 15(68.2) | 6(27.3) | 3432.5(2127–11541) |
| CTGCTTCT | 18 (5.8) | 1(5.6) | 8(44.4) | 9(50) | 2427.5(1679–3700) |
| CTGTTTCT | 6 (1.9) | 1(16.7) | 3(50) | 2(33.3) | 2619(1868–5909) |
| CGACTCCT | 5 (1.5) | 1(20) | 4(80) | 0 | 3424(2592–3661) |
| TGATTCCC | 3 (1) | 0 | 3(100) | 0 | 3067(1784–5139) |
| TGACTCCT | 2 (0.6) | 0 | 1(50) | 1(50) | 2805(2517–3093) |
| TGATTCCT | 2 (0.6) | 0 | 2(100) | 0 | 3514.5(2765–4264) |
| TTGCTTCT | 2 (0.6) | 0 | 0 | 2(100) | 4464(1422–7506) |
| CGACCCCT | 2 (0.6) | 0 | 0 | 2 (100) | 2697(2237–3157) |
| CGGCTTCC | 1 (0.3) | 0 | 1(100) | 0 | 3695 (3695–3695) |
| TGATTTCC | 1 (0.3) | 0 | 1(100) | 0 | 3727(3727–3727) |

$p < 0.039$), T allele of CYP2B6 516G>T ($\beta$ = 1868.9, 95% CI 3216.9–3416.3; $p < 0.018$), the C allele of CYP2B6 983T>C ($\beta$ = 2638.3, 95% CI = 1348–3929; $p < 0.0001$), T allele of CYP2B6 21563C>T ($\beta$ = 1737, 95% CI = 972.2–2681.9; $p < 0.0001$) and the presence of 5 to 7 numbers of SNPs per patient ($\beta$ = 570, 95% CI = 362–778; $p < 0.0001$). Having HIV viral load ≤1000 cells/mL was associated with lower EFV plasma levels ($\beta$ = -4199.3, 95% CI = -7914.9 –-483.6; $p = 0.027$). A trend was also observed for non-adherence in the past 30 days ($\beta$ = -419.1, 95% CI = -916–77.9; $p = 0.098$) in association with lower EFV plasma levels (Table 5).

## Discussion

This is among the first studies with sufficient samples size reporting the association between efavirenz plasma concentrations and expanded CYP2B6 genetic variants and one CAR in one of the largest cosmopolitan ARV treatment centers in Nairobi Kenya. Although majority of patients' EFV plasma concentrations (63.8%) were within the therapeutic window, 4.5% of the patients had suboptimal EFV concentrations. This outcome is significant given EFV is the only NNRTI still retained as part of first line ART treatment in Kenya [24]. Further, these results are significant given that the use of EFV is associated with treatment discontinuation due to neurotoxicity [25] implying an existing challenge in the management of HIV patients receiving this regimen in Kenya.

The acceptable minimum EFV plasma concentration required to attain virologic suppression is indicated as >1000 ng/mL [26]. Sub-optimal EFV plasma levels reduces viral suppression and is associated with the development ART resistant mutations [20,27,28]. In agreement with our study, about 33.3% of the patients with virologic failure also had suboptimal EFV plasma concentration. The presence of HIV resistant mutation was associated with lower EFV plasma levels. Thirty percent (30%) of the 10 patients with drug resistant mutation had suboptimal plasma level with some 10% having EFV plasma level considered supra-therapeutic levels. The dynamic nature in the frequency of specific HIV drug resistant mutations conferring resistance to NNRTI including (K103N, Y181C and G190A) as a results of long-term

**Table 5. Quartile regression analyses of factors associated with EFV plasma concentrations.**

| Variable | Univariate analysis | | | Multivariate analysis | | |
|---|---|---|---|---|---|---|
| | Unadjusted β | (95% CI) | P-value | Adjusted β | (95% CI) | P-value |
| Age | -14 | -38.1 | 10.8 | 0.273 | -7.4 | -33.7 | 19 | 0.582 |
| Gender | -35 | -545.9 | 475.9 | 0.893 | -325.8 | -925 | 273.3 | 0.285 |
| Alcohol use | 330 | -296.1 | 956.1 | 0.3 | 293.5 | -632.2 | 1219.2 | 0.533 |
| Smoking | -715 | -2413.9 | 983.9 | 0.408 | -55.7 | -1589.6 | 1478.2 | 0.943 |
| Skin rash | 788 | 114.2 | 1461.8 | 0.022 | 1379 | 72.5 | 2685.5 | 0.039 |
| ART regimen | 318 | -148.5 | 784.5 | 0.181 | 316.6 | -326.2 | 959.4 | 0.333 |
| None-adherence | -400 | -826.7 | 26.7 | 0.066 | -419.1 | -916 | 77.9 | 0.098 |
| Months since ART initiation | -2.3 | -16.1 | 11.5 | 0.743 | -8.6 | -35.2 | 18 | 0.526 |
| HIV drug resistant mutation | 1388 | 304.8 | 2471.2 | 0.012 | 1314.5 | -937.4 | 3566.3 | 0.252 |
| HIV-RNA <1000 cps/mL | -1390 | -2676.2 | -103.8 | 0.034 | -4199.3 | -7914.9 | -483.6 | 0.027 |
| CD4 count | 0.04 | -1.5 | 1.6 | 0.953 | 0.4 | -0.2 | 1.1 | 0.18 |
| AST | 1.9 | -4.6 | 8.4 | 0.558 | 0.1 | -7.4 | 7.5 | 0.984 |
| ALT | 6.96 | -2.4 | 16.3 | 0.144 | 7.1 | -10.4 | 24.6 | 0.427 |
| BMI | -26.9 | -90.7 | 36.9 | 0.408 | -11.4 | -74.4 | 51.6 | 0.722 |
| 15582C>T | 54 | -422.9 | 530.9 | 0.824 | 293.7 | -727.6 | 1315.1 | 0.572 |
| 516G>T | 1583 | 1137.1 | 2028.9 | 0.0001 | 1868.9 | 321.6 | 3416.3 | 0.018 |
| 785A>G | 1570 | 1061 | 2079 | 0.0001 | 711 | -9053.1 | 10475.1 | 0.886 |
| 18492C>T | -896 | -1398.5 | -393.5 | 0.001 | -444.4 | -1360.7 | 471.9 | 0.341 |
| 983T>C | 2068 | 467.9 | 3668.1 | 0.02 | 2638.3 | 1348 | 3929 | 0.0001 |
| 21563C>T | 1577 | 1083.5 | 2070.5 | 0.0001 | 1737 | 792.2 | 2681.8 | 0.0001 |
| 1459C>T/A | 364 | -267177 | 266449 | 0.998 | 135 | -318857 | 319126.6 | 0.999 |
| 540C>T | 275 | -559.5 | 1109.5 | 0.542 | 230.5 | -817.3 | 1278.4 | 0.665 |
| Number of SNPs per patient | 517 | 364 | 670 | 0.0001 | 570 | 362 | 778 | 0.0001 |
| Haplotypes | **107** | -26.1 | 240.1 | 0.115 | 16.4 | -64.6 | 97.4 | 0.69 |

ART treatment play a big role in the treatment outcome [29]. Inevitably, drug resistant mutations whether transmitted, acquired or archived are crucial in determining the treatment outcomes especially for the NNRTIs which have been shown to have a relatively low genetic barrier to resistance [30].

Though we reported no association between EFV with hepatotoxicity (ALT and AST), there was evidence towards increased ALT and AST levels among patients with supra-therapeutic EFV plasma concentrations. Studies have associated patients with supra-therapeutic plasma NNRTI, with a high risk of developing drug toxicity, a common cause of ART treatment non- adherence and discontinuation medications [31].

The prevalence of variant alleles for CYP2B6 516T, 983C, CYP2B6 18492T, CYP2B6 21563T, CYP2B6 1459T and CYP2B6 785G in this population was similar to that reported for other African populations and Kenyan ethnic groups [32–34]. For the CAR 540T C>T, the frequency of the T variant allele was 14.9% which was similar to those reported by Wyen, *et al* [35] which showed 15% variant alleles in black population. The allele frequencies of 516G>T and 785A>G have been consistently higher in the Asian populations, such as Chinese populations [36], Japanese and Korean [37], but lower than that in African populations [14,38]. Further, the allele frequencies of CYP2B6 983C was 14.1% which is consist to other studies showing predominant occurrence of this allele in African subjects [38,39]. Studies have demonstrated the role of polymorphisms in the CYP2B6 gene on enzymatic activity with concomitant effect on EFV concentrations [36,40]. In this study were reported an increased EFV

plasma level with the presence of the T and C allele at the position c.516 and of 983 CYP2B6 respectively in agreement with previous studies [33,36,40]. For CYP2B6 983T>C genotype, both the homozygosity and heterozygosity for mutant were associated with higher EFV plasma concentration than those patients with wild-type 983CC. The effect was noteworthy for its magnitude [41]. The presence of T allele at the position c.18492 of CYP2B6 18492 was associated lower plasma efavirenz concentrations, which is consistent with previous studies [34,40,42]. Studies have shown the presence of this CYP2B6 18492T>C SNP together with coadministration of a strong CYP inducer may increase the likelihood of subtherapeutic plasma efavirenz concentrations [34].

This is the first report in Kenya to assess the role of constitutive androstane receptor (CAR) 540T C>T in the concentrations for EFV. The frequency of the T variant allele for CAR 540 C>T was 12.2% in agreement, Wyen et al., [35] reported the prevalence of T variant in 32.7% among the Caucasians and 15% among the blacks. In Ghana Sarfo et al., [43] reported slightly lower the frequency of variant allele for CAR 540C>T at 7% among patients. In our study, though not significant in linear regression analysis, the homozygous and heterozygous for mutation for CAR 540T C>T had higher EFV plasma concentrations. This was consistent with the two reports which observed a trend towards association between plasma efavirenz concentration and CAR 540C>T [35,43].

Haplotype analysis evaluates the interactions of multiple SNPs, leading to a decrease or increase in the metabolic function of CYP2B6 or CAR. Haplotype might accurately predict ARV drug pharmacokinetics than a single SNP [16]. In this study, linkage disequilibrium among 15582C>T, 516G>T, 785A>G, 18492C>T, 983T>C, 21563C>T, 1459C>T and CAR 540C>T was observed, resulting in 16 haplotypes among which CTGCTTCC and CGGCTTCC or TGATTTCC had the highest and the lowest frequency, respectively. Compared to CTGCTTCC, 4 haplotypes (CTGCTTCC, CTGCTTCT, TTGCTTCT and CGACCCCT) were associated with higher EFV plasma concentration). Meng et al., (2015) [36] showed the predictive accuracy of the average EFV plasma concentration was higher among the haplotypes than with each single SNPs.

This study has three major limitations. First, the cross-sectional nature of this study had the potential of introducing confounding factors as well as only permitting describing the relationship between EFV plasma concentrations, patient genetics and a few pharmacoecologic factors and not a causal conclusion. Such outcomes can be confirmed in a longitudinal study. Second, although we excluded patients on tuberculosis, hepatitis B or C virus co-infection was not tested during the study. The prevalence of hepatitis B or C virus coinfection is generally high among HIV patients [44,45]. Studies have reported the influence of HCV co-infection on nevirapine plasma levels [28], with the effect of this HCV coinfection being minimal among patient with a normal liver function [46]. Hepatitis co-infection and potential influence on EFV plasma levels particularly among patients with hepatotoxicity cannot be ruled out in this study. Third, even though the study was conducted in a cosmopolitan treatment center, the outcome from this geographically define subset of patients may not be generalized to other patients majorly because ethnicity and environmental factors influences variations in drug levels. One of the strengths of this study was that, blood samples were collected among patients who had been on ARV treatment for 12 months. Further, sample collections were done between 12 and 16 h after EFV administration. These two factors might have mitigated the effects of non-compliance and inter-individual variability [47].

Given these limitations, the following conclusions can be drawn from these data. Broader interindividual variability in efavirenz plasma concentrations was reported with a relatively large percentage (36.2%) of the patients having EFV plasma concentrations beyond therapeutic window, posing high risk of treatment failure or toxicity. Other than *CYP2B6* c.516G>T

and CYP2B6 983T>C polymorphism, four *CYP2B6* gene (785A>G, 18492C>T, 21563C>T and 15582C>T) and one CAR (540T>C) are potential predictors of efavirenz plasma levels Haplotype analysis suggested strong association of `CTGCTTCC`, `CTGCTTCT`, `TTGCTTCT` and `CGACCCCT` and EFV plasma concentration.

## Supporting information

**S1 Fig. The differences in log$_{10}$-transformed EFV plasma concentrations by genotypes of 7 CYP2B6 and 1 CAR SNPs.** 15582C>T, 516G>T, 785A>G, 983T>C, 21563C>T and 18492C>T significantly influence EFV plasma concentration ($p < 0.05$) but not 1459C>T and CAR 540C>T.
(PDF)

**S2 Fig. Linkage disequilibrium analysis of 7 SNPs of CYP2B6 and 1 CAR.** Dark red squares: strong evidence of LD, dark yellow/orange squares: uninformative, light yellow squares: strong evidence of recombination. SNP1-15582C>T; SNP2 - 516G>T; SNP3 - 785A>G; SNP4- 18492C>T; SNP5- 983T>C; SNP6-21563C>T; SNP7- 1459C>T and SNP8—CAR 540C>T.
(PDF)

## Acknowledgments

We wish to acknowledge all the study patients and staff of FACES-KEMRI program, the Director KEMRI, and all the staff of the Retrovirology, Centre de Recherche Public de la Sant, Luxembourg.

## Author Contributions

**Conceptualization:** Musa Otieno Ngayo, Zachari Arochi Kwena, Wallace Dimbuson Bulimo.

**Data curation:** Musa Otieno Ngayo.

**Formal analysis:** Musa Otieno Ngayo, Faith Apolot Okalebo.

**Funding acquisition:** Musa Otieno Ngayo.

**Investigation:** Margaret Oluka.

**Methodology:** Musa Otieno Ngayo, Margaret Oluka, Zachari Arochi Kwena, Wallace Dimbuson Bulimo, Faith Apolot Okalebo.

**Supervision:** Margaret Oluka, Wallace Dimbuson Bulimo, Faith Apolot Okalebo.

**Validation:** Margaret Oluka.

**Writing – original draft:** Musa Otieno Ngayo.

**Writing – review & editing:** Margaret Oluka, Zachari Arochi Kwena, Wallace Dimbuson Bulimo, Faith Apolot Okalebo.

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
