## [Decision Letter · Decision Letter 0]

1 Mar 2021

PONE-D-21-00146

Effects of cytochrome P450 2B6 and constitutive androstane receptor genetic variation on Efavirenz plasma concentrations among HIV patients in Kenya

PLOS ONE

Dear Dr. Ngayo,

Thank you for submitting your manuscript to PLOS ONE. After careful consideration, we feel that it has merit but does not fully meet PLOS ONE’s publication criteria as it currently stands. Therefore, we invite you to submit a revised version of the manuscript that addresses the points raised during the review process.

In particular, points have been to be improved in bringing precisions in the material and methods, throughout the whole text, and clarifications in figures and results should be done.

We look forward to receiving your revised manuscript.

Kind regards,

Isabelle Chemin, PhD

Academic Editor

PLOS ONE

Journal Requirements:

2. Thank you for submitting the above manuscript to PLOS ONE. During our internal evaluation of the manuscript, we found significant text overlap between your submission and the following previously published works, some of which you are an author.

https://www.futuremedicine.com/doi/10.2217/pgs.11.160 (Introduction, paragraph 1, sentences 1-2)

https://www.zora.uzh.ch/id/eprint/56011/1/Efavirenz-CYP2B6-CAR-JAntimicrobChemother2011-ZORA.pdf (Introduction, paragraph 1, sentences 4-5)

https://academic.oup.com/jac/article/66/9/2092/771232  (Introduction, paragraph 1, sentence 6 & paragraph 2, sentence 1)

https://pericles.pericles-prod.literatumonline.com/doi/10.1002/mgg3.598 (Introduction, paragraph 2, sentence 2)

https://journals.plos.org/plosone/article?id=10.1371%2Fjournal.pone.0130583 (Introduction, paragraph 2, sentences 3-4 & Discussion, paragraph 2, sentence 3)

https://journals.plos.org/plosone/article?id=10.1371/journal.pone.0130583 (Discussion, paragraph 1, sentence 1)

https://journals.sagepub.com/doi/10.1177/2050312118780861 (Discussion, paragraph 4, sentence 2)

https://f1000research.com/articles/9-363 (Discussion, paragraph 6, sentence 2 & paragraph 10, sentence 2)

https://ascpt.onlinelibrary.wiley.com/doi/full/10.1002/cpt.1477 (Discussion, paragraph 6, sentence 2)

https://www.nature.com/articles/s41598-018-23350-1 (Discussion, paragraph 11, sentence 3)

https://insights.ovid.com/british-clinical-pharmacology/bjcpha/2012/12/000/pharmacogenetic-markers-cyp2b6-associated/13/00002256 (Discussion, paragraph 11, sentence 4)

Please revise the manuscript to rephrase the duplicated text, cite your sources, and provide details as to how the current manuscript advances on previous work. Please note that further consideration is dependent on the submission of a manuscript that addresses these concerns about the overlap in text with published work.

We will carefully review your manuscript upon resubmission, so please ensure that your revision is thorough

4. Please ensure that you refer to Figure 2 in your text and check your figure legend numbering in the manuscript as, if accepted, production will need this reference to link the reader to each figure.

Reviewers' comments:

Reviewer's Responses to Questions

**Comments to the Author**

1. Is the manuscript technically sound, and do the data support the conclusions?

Reviewer #1: Yes

2. Has the statistical analysis been performed appropriately and rigorously? 

Reviewer #1: No

3. Have the authors made all data underlying the findings in their manuscript fully available?

Reviewer #1: Yes

4. Is the manuscript presented in an intelligible fashion and written in standard English?

Reviewer #1: No

5. Review Comments to the Author

Reviewer #1: The authors report associations of genetic variation of CYP2B6 (13 variants) and constitutive androstane receptor (CAR) (one variant) on efavirenz (EFV) plasma concentration among 312 HIV Kenyan patients who were on efavirenz-based first-line ART regimen. The authors confirm published associations between CYP2B6 variants and efavirenz exposure.

Major comments

• The manuscript is poorly communicated - needs a careful editing of several typos, repetitive statements and inaccurate references

• The way the data has been analyzed is somewhat inadequate (see below) and it needs careful check by a statistician. Moreover, it would be advisable to go over the tables and figures carefully to make sure the values presented are correct and make sense.

• The basis for the calculation of metabolic score is not well justified and the data derived from it is suspect.

• Too much redundancy in the discussion section and this part could be significantly shortened without losing the major findings.

Other comments

Abstract:

1. What does (p>0.0001) mean?

2. There seems a discrepancy in the proportion of patients outside the efavirenz concentration (22% versus 36.2%). Please correct this discrepancy.

3. How was “High number of patients (17.9%) had an inferred ultra-rapid metabolic phenotype” determined.

Introduction:

4. First paragraph (2nd sentence) has multiple problems. First, Wyen et al., 2013 is given as a reference for CYP2A6-mediated efavirenz metabolism, which was not included in the reference list. Second, I am not sure that is the best reference for CYP2A6-mediated metabolism (PMID: 20335270 probably better). Third, I am not sure why CYP2B6 is repeated in that sentence. I think, that sentence needs revision by saying for example such as….Efavirenz 7-hydroxylation by CYP2B6 and direct N-glucuronidation by UGT2B7 represent minor metabolic pathways. Please use the original references for CYP2A6 and UGT2B7.

5. First paragraph: UGT2B is probably type and shows up out of the blue? I would imagine this to be UGT2B7 and that is why they need first to provide metabolism by UGT2B7 as suggested above.

6. First paragraph: Wyen et al., 2013 is given in the text, but Wyen et al., 2011 is given in the reference list. Please address this inconsistency

7. Second paragraph: CYP2B6 allele nomenclature. www.cypalleles.ki.se/cyp2b has been retired. Use the PharmVar web page for this reference

Study design and settings:

8. The citation Ngayo et al., (2016) is not listed in the reference list

Data collection:

9. Was the whole blood sample for genotyping and drug bassay collected in the same tube or different tubes? Please specify

LC/MS/MS assay:

10. Provide assay dynamics (intra- and interday, LOQ, linear range etc). Also what was the internal standard used

Results section:

11. Table 2: It is intriguing that the HIV-RNA >1000copies/mL and mutations was more prominent in the normal efavirenz concentration range than the lower range. It is stated that non-adherence rate was 23.1%. Was this taken into account among the low, normal and high efavirenz concentration group? It may also help to include the non-adherence rate among the groups in Table 2.

12. What was the basis for calculating the metabolic scores?. This calculation seems to have major flaws. First, it assumes the functional consequences of each SNP are the same. This assumption is wrong. In terms of in vivo functional consequences, the SNPs can be categorized as follows from high impact to low impact: CYP2B6 983T>C (almost null allele) > CYP2B6 516G>T >> CYP2B6 15582C>T and CYP2B6 1459C>T (functional). CYP2B6 785A>G is in linkage disequilibrium with CYP2B6 516G>T forming CYP2B6*6, but when it occur alone (*4) is associated with increased activity. Although CYP2B6 18492C>T is associated with lower efavirenz concentrations, how is that possible to assign 18492 TT = −2; 18492 CT = −1; 18492 CC = 0. I think this type of functional classification and the subsequent designation of metabolic status (Figure 2) and results associated is unlikely to be valid. How inferred ultra-rapid is also assigned remains unclear. Overall, this paragraph is very confusing. There are also typos (1459TT, 1459TC and 1459TT).

13. Table 4 and the texts in the body of the manuscript. The 95 % CI are the same as the difference. Something wrong?

14. The values listed in Table 5, particularly the Beta Coefficients, seem odd. First, the direction of effect in the unadjusted and adjusted does not always match (e.g., unadjusted beta for 21563C>T 1577 while adjusted is -8512.8). Second, the values of the beta are huge and unrealistic. It may be prudent to check the statistical analysis.

Discussions:

15. First paragraph (1st and 2nd sentence): the link between CYP2B6 genetics and efavirenz concentrations has been established in many populations of African descent. I would start with a sentence something like “This is the first analysis in Kenyans with sufficient sample size and expanded variants in CYP2B6 variants, etc”. Also it could not be comprehensive for CAR genetics when only one SNP was tested.

16. 2nd paragraph: provide appropriate references to the last sentence

17. Third paragraph (about metabolic score): remove this paragraph and Figure 1 unless providing adequate justification for the categorization (see comments #9 above).

18. Fourth paragraph: the authors state that “In agreement with our study, a significant proportion of patients with virologic failure (33.3%) had suboptimal EFV plasma concentration compared to 10% of the patients with EFV plasma concentration within therapeutic range”. I do not think this sentence is consistent with findings in Table 2 where the prevalence is much higher in the group with normal range

19. Fifth paragraph. Were patients with TB and on anti-TB included in this study, which might have created the discrepancy with the other studies?

20. Six paragraph: the lack of PK interaction between efavirenz and NRIs has been well established and does not add new information (can be deleted).

21. Typo (two figure 1). I think Figure 1 can go to supplemental files (data are already displayed in Table 2). The other figure 1 (which should be figure 2) could be deleted.

22. Figure 3 is difficult to read and can go to supplemental file.

6. PLOS authors have the option to publish the peer review history of their article (what does this mean?). If published, this will include your full peer review and any attached files.

Reviewer #1: No

---

## [Author Response · Author response to Decision Letter 0]

20 Jul 2021

Isabelle Chemin, PhD

Academic Editor

PLOS ONE

RE: Response to reviewers’ comments for the manuscript PONE-D-21-00146

Effects of cytochrome P450 2B6 and constitutive androstane receptor genetic variation on Efavirenz plasma concentrations among HIV patients in Kenya

Find a point by point responses to the issued raised by reviewers

Comment: Improved in bringing precisions in the material and methods, throughout the whole text, and clarifications in figures and results should be done.

Response: The methodology has been improved and the protocol used also deposited in the laboratory protocols in protocols.io to enhance the reproducibility of your results. 

Comment: Please ensure that your manuscript meets PLOS ONE's style requirements, including those for file naming. 

Response: The format of the manuscript has been adjusted in line with the PLOS ONE style templates can be found at

Comment: During our internal evaluation of the manuscript, we found significant text overlap between your submission and the following previously published works, some of which you are an author.

https://www.futuremedicine.com/doi/10.2217/pgs.11.160 (Introduction, paragraph 1, sentences 1-2)

https://www.zora.uzh.ch/id/eprint/56011/1/Efavirenz-CYP2B6-CAR-JAntimicrobChemother2011-ZORA.pdf (Introduction, paragraph 1, sentences 4-5)

https://academic.oup.com/jac/article/66/9/2092/771232 (Introduction, paragraph 1, sentence 6 & paragraph 2, sentence 1)

https://pericles.pericles-prod.literatumonline.com/doi/10.1002/mgg3.598 (Introduction, paragraph 2, sentence 2)

https://journals.plos.org/plosone/article?id=10.1371%2Fjournal.pone.0130583 (Introduction, paragraph 2, sentences 3-4 & Discussion, paragraph 2, sentence 3)

https://journals.plos.org/plosone/article?id=10.1371/journal.pone.0130583 (Discussion, paragraph 1, sentence 1)

https://journals.sagepub.com/doi/10.1177/2050312118780861 (Discussion, paragraph 4, sentence 2)

https://f1000research.com/articles/9-363 (Discussion, paragraph 6, sentence 2 & paragraph 10, sentence 2)

https://ascpt.onlinelibrary.wiley.com/doi/full/10.1002/cpt.1477 (Discussion, paragraph 6, sentence 2)

https://www.nature.com/articles/s41598-018-23350-1 (Discussion, paragraph 11, sentence 3)

https://insights.ovid.com/british-clinical-pharmacology/bjcpha/2012/12/000/pharmacogenetic-markers-cyp2b6-associated/13/00002256 (Discussion, paragraph 11, sentence 4)

Response: in the revised manuscript all the duplicated text have been rephrased. The revised manuscripts also show the advances in knowledge from previous work

Comment: In your Data Availability statement, you have not specified where the minimal data set underlying the results described in your manuscript can be found. PLOS defines a study's minimal data set as the underlying data used to reach the conclusions drawn in the manuscript and any additional data required to replicate the reported study findings in their entirety. All PLOS journals require that the minimal data set be made fully available. For more information about our data policy, please see http://journals.plos.org/plosone/s/data-availability.

Response: Data used in this study have been supplied as Supporting Information files

Comment. Please ensure that you refer to Figure 2 in your text and check your figure legend numbering in the manuscript as, if accepted, production will need this reference to link the reader to each figure.

 Response: The figure and table numberings and reference in the text have been redone in line with manuscript guidelines

Comment: Your ethics statement should only appear in the Methods section of your manuscript. If your ethics statement is written in any section besides the Methods, please delete it from any other section.

Response: The ethics statement has been inserted in the Methods section only

Reviewers' comments

Reviewer's Responses to Questions

Comments to the Author

1. Is the manuscript technically sound, and do the data support the conclusions?

Reviewer #1: Yes

2. Has the statistical analysis been performed appropriately and rigorously? 

Reviewer #1: No

3. Have the authors made all data underlying the findings in their manuscript fully available?

Reviewer #1: Yes

4. Is the manuscript presented in an intelligible fashion and written in standard English?

Reviewer #1: No

5. Review Comments to the Author

Reviewer #1: The authors report associations of genetic variation of CYP2B6 (13 variants) and constitutive androstane receptor (CAR) (one variant) on efavirenz (EFV) plasma concentration among 312 HIV Kenyan patients who were on efavirenz-based first-line ART regimen. The authors confirm published associations between CYP2B6 variants and efavirenz exposure.

Major comments

Comment: The manuscript is poorly communicated - needs a careful editing of several typos, repetitive statements and inaccurate references

Response: The manuscript has been edited to improve on language and eliminate typos, repetitive statement and inaccurate references

Comment: The way the data has been analyzed is somewhat inadequate (see below) and it needs careful check by a statistician. Moreover, it would be advisable to go over the tables and figures carefully to make sure the values presented are correct and make sense.

Response: Data re-analyzed and tables and figures rechecked and improved and or corrected.

Comment: The basis for the calculation of metabolic score is not well justified and the data derived from it is suspect.

Response: The whole section of metabolic score has been dropped from the manuscript

Comment: Too much redundancy in the discussion section and this part could be significantly shortened without losing the major findings.

Response: Discussion section edited appropriately to improve on the flow

Other comments

Abstract:

Comment 1. What does (p>0.0001) mean?

Response. The p value has been corrected

Comment 2. There seems a discrepancy in the proportion of patients outside the efavirenz concentration (22% versus 36.2%). Please correct this discrepancy.

Response. Discrepancy corrected

Comment 3. How was “High number of patients (17.9%) had an inferred ultra-rapid metabolic phenotype” determined.

Response: Metabolic section deleted from the document

Introduction:

Comments 4. First paragraph (2nd sentence) has multiple problems. First, Wyen et al., 2013 is given as a reference for CYP2A6-mediated efavirenz metabolism, which was not included in the reference list. Second, I am not sure that is the best reference for CYP2A6-mediated metabolism (PMID: 20335270 probably better). Third, I am not sure why CYP2B6 is repeated in that sentence. I think, that sentence needs revision by saying for example such as…. Efavirenz 7-hydroxylation by CYP2B6 and direct N-glucuronidation by UGT2B7 represent minor metabolic pathways. Please use the original references for CYP2A6 and UGT2B7.

5. First paragraph: UGT2B is probably type and shows up out of the blue? I would imagine this to be UGT2B7 and that is why they need first to provide metabolism by UGT2B7 as suggested above.

6. First paragraph: Wyen et al., 2013 is given in the text, but Wyen et al., 2011 is given in the reference list. Please address this inconsistency

7. Second paragraph: CYP2B6 allele nomenclature. www.cypalleles.ki.se/cyp2b has been retired. Use the PharmVar web page for this reference

Repose: The introduction has been rewritten to eliminate reference mismatch and those missing. All the original references have been included and those missing included. The inconsistency in references also addressed. The reference listing CYP2B6 alleles has been changed from CYP2B6 allele nomenclature. www.cypalleles.ki.se/cyp2b to Pharmacogene Variation Consortium. Available at https://www.pharmvar.org/gene/CYP2B6 accessed July, 2021

Study design and settings:

Comment 8. The citation Ngayo et al., (2016) is not listed in the reference list.

Response: This reference was cited as number 24 in the reference list except that the author’s full names were written as Musa Otieno Ngayo. This has since been adjusted accordingly

Data collection:

Comment 9. Was the whole blood sample for genotyping and drug bassay collected in the same tube or different tubes? Please specify

Response: The blood collection tubes have been specified as follows “Blood samples (10 mL) at 12–16 h post ARV uptake was collected into three blood tube as follows: EDTA anticoagulant tube for immunological testing and CYP2B6 and CAR genotyping. Serum separating tube for clinical chemistry while Lithium heparin tube for HIV viral load and EFV plasma level quantification

LC/MS/MS assay:

Comment 10. Provide assay dynamics (intra- and interday, LOQ, linear range etc). Also what was the internal standard used

Response: the assay dynamics have been presented as solutions, selectivity, Recovery and Linearity and accuracy and precision in the method section

Results section:

Comment 11a. Table 2: It is intriguing that the HIV-RNA >1000copies/mL and mutations was more prominent in the normal efavirenz concentration range than the lower range. 

Response. The distribution of viral load remains as stated above. Data has been provided for confirmation

Comment 11b It is stated that non-adherence rate was 23.1%. Was this taken into account among the low, normal and high efavirenz concentration group? It may also help to include the non-adherence rate among the groups in Table 2.

Response. The non-adherence rate for the past 30 days was reported as 207/312 (66.3) which has been compared across the EFV therapeutic ranges. This has been included in the table 2

Comment 12. What was the basis for calculating the metabolic scores? This calculation seems to have major flaws. First, it assumes the functional consequences of each SNP are the same. This assumption is wrong. In terms of in vivo functional consequences, the SNPs can be categorized as follows from high impact to low impact: CYP2B6 983T>C (almost null allele) > CYP2B6 516G>T >> CYP2B6 15582C>T and CYP2B6 1459C>T (functional). CYP2B6 785A>G is in linkage disequilibrium with CYP2B6 516G>T forming CYP2B6*6, but when it occur alone (*4) is associated with increased activity. Although CYP2B6 18492C>T is associated with lower efavirenz concentrations, how is that possible to assign 18492 TT = −2; 18492 CT = −1; 18492 CC = 0. I think this type of functional classification and the subsequent designation of metabolic status (Figure 2) and results associated is unlikely to be valid. How inferred ultra-rapid is also assigned remains unclear. Overall, this paragraph is very confusing. There are also typos (1459TT, 1459TC and 1459TT).

Response: All the section of Metabolic score and inferred phonotypes has been deleted from the manuscript

Comment 13. Table 4 and the texts in the body of the manuscript. The 95 % CI are the same as the difference. Something wrong?

Response: The data has been re-analyzed by a statistician and discrepancies corrected both in the text and in Table

Comment 14. The values listed in Table 5, particularly the Beta Coefficients, seem odd. First, the direction of effect in the unadjusted and adjusted does not always match (e.g., unadjusted beta for 21563C>T 1577 while adjusted is -8512.8). Second, the values of the beta are huge and unrealistic. It may be prudent to check the statistical analysis.

Response: The data has been re-analyzed by a statistician and discrepancies corrected both in the text and in Table

Discussions:

Comment 15. First paragraph (1st and 2nd sentence): the link between CYP2B6 genetics and efavirenz concentrations has been established in many populations of African descent. I would start with a sentence something like “This is the first analysis in Kenyans with sufficient sample size and expanded variants in CYP2B6 variants, etc”. Also it could not be comprehensive for CAR genetics when only one SNP was tested.

Response: The sentence has been edited to reflect the true picture

Comment 16. 2nd paragraph: provide appropriate references to the last sentence

Response: Reference by Haas ety al., 2004 cited

Comment 17. Third paragraph (about metabolic score): remove this paragraph and Figure 1 unless providing adequate justification for the categorization (see comments #9 above).

Comment 18. Fourth paragraph: the authors state that “In agreement with our study, a significant proportion of patients with virologic failure (33.3%) had suboptimal EFV plasma concentration compared to 10% of the patients with EFV plasma concentration within therapeutic range”. I do not think this sentence is consistent with findings in Table 2 where the prevalence is much higher in the group with normal range

Response; The sentence corrected in line with the table

Comment 19. Fifth paragraph. Were patients with TB and on anti-TB included in this study, which might have created the discrepancy with the other studies?

Response; TB patients or those on anti-TB not included in the study

Comment 20. Six paragraph: the lack of PK interaction between efavirenz and NRIs has been well established and does not add new information (can be deleted).

Response; This paragraph deleted

Comment 21. Typo (two figure 1). I think Figure 1 can go to supplemental files (data are already displayed in Table 2). The other figure 1 (which should be figure 2) could be deleted.

Response. The figures corrected accordingly and figure 1included in the supplementary section 

Comment 22. Figure 3 is difficult to read and can go to supplemental file.

Response. Figure 3 taken to supplementary section as S2 Fig

6. PLOS authors have the option to publish the peer review history of their article (what does this mean?). If published, this will include your full peer review and any attached files.

Do you want your identity to be public for this peer review? For information about this choice, including consent withdrawal, please see our Privacy Policy.

Thank you for your valuable comments which we have incorporated the suggested changes and we hope that manuscript meets the requirements of the journal.

Looking forward for a positive response from you

Kind regards,

Dr. Musa Otieno Ngayo

Centre for Microbiology Research

KEMRI

Enclosed is our manuscript

---

## [Editor Report · Decision Letter 1]

19 Nov 2021

Effects of cytochrome P450 2B6 and constitutive androstane receptor genetic variation on Efavirenz plasma concentrations among HIV patients in Kenya

PONE-D-21-00146R1

Dear Dr. Ngayo,

We’re pleased to inform you that your manuscript has been judged scientifically suitable for publication and will be formally accepted for publication once it meets all outstanding technical requirements.

Kind regards,

Isabelle Chemin, PhD

Academic Editor

PLOS ONE

Additional Editor Comments (optional):

The comments raised during the review process on several aspects were well answered in a comprehensive manner.
---

## [Editor Report · Acceptance letter]

2 Feb 2022

PONE-D-21-00146R1 

Effects of cytochrome P450 2B6 and constitutive androstane receptor genetic variation on Efavirenz plasma concentrations among HIV patients in Kenya 

Dear Dr. Ngayo:

I'm pleased to inform you that your manuscript has been deemed suitable for publication in PLOS ONE. Congratulations! Your manuscript is now with our production department. 

Kind regards, 

on behalf of

Mrs Isabelle Chemin 

Academic Editor

PLOS ONE